# Time-Dependent Effects of Physical Activity on Cardiovascular Risk Factors in Adults: A Systematic Review

**DOI:** 10.3390/ijerph192114194

**Published:** 2022-10-30

**Authors:** Hengxu Liu, Shiqi Liu, Kun Wang, Tingran Zhang, Lian Yin, Jiaqi Liang, Yi Yang, Jiong Luo

**Affiliations:** Research Centre for Exercise Detoxification, College of Physical Education, Southwest University, Chongqing 400715, China

**Keywords:** circadian rhythm, exercise, cardiovascular, chronotype, review

## Abstract

Purpose: Physical activity is an important non-drug-related method to prevent and treat cardiovascular diseases, but how exercise duration affects the cardiovascular metabolic risk factors in adults remains uncertain. This review systematically examines the time-dependent effects of physical activity on cardiovascular risk factors in adults and aims to further the understanding of the temporal therapeutics of exercise. Methods: Following the PRISMA guidelines, the PubMed, Web of Science, EMBASE, and CNKI databases were systematically searched for relevant scientific studies from January 2000 to June 2022. Results: A total of 16 studies met the inclusion criteria and were included in the systematic review. The sample size ranged from 11–275 participants who were diagnosed with obesity, hypertension, diabetes mellitus type 2 (T2DM), and Coronary Heart Disease (CAD), while the subjects in four studies did not report any metabolic or cardiovascular disease. Four studies conducted trials of acute exercise interventions, while the remaining intervention periods ranged from 12 days to 12 weeks. The exercise interventions included aerobic training, resistance training, aerobic training that was combined with resistance training, compound exercise, and high-intensity interval exercise, and the training frequency varied from 2–5 times/week. Conclusions: Overall, this review found some evidence that the cardiovascular risk factors in adults may be time-dependent in response to physical activity. However, it is limited by the small sample size for each of the outcomes and several methodological issues, leading to poor comparability between studies. A randomized controlled trial with a larger sample size is supposed to be designed for the relevant population to completely test whether synchronizing the exercise time point in the day with the individual’s circadian rhythm can amplify the benefits of the exercise for improving cardiovascular health.

## 1. Introduction

Cardiovascular diseases (CVD) have become one of the major threats to human public health worldwide [1], with them ranking first in the global disease burden. The occurrence and development of CVD are the result of the interaction of multiple adverse factors. Epidemiological studies showed that more than 40% of CVD can be attributed to metabolic risk factors, obesity, and abnormalities in blood lipids, glucose and electrolytes which can damage the anatomy and physiology of the heart and blood vessels, and the superposition of multiple disorders factors is positively correlated with the risk of CVD [2]. In addition, the inflammatory state of the circulatory system is also of great value in predicting cardiovascular events because the body is prone to CVD when it is in a state of chronic systemic inflammation. TNF-A, IL-6, and CRP are the most widely investigated inflammatory factors that are related to CVD risk. C-reactive protein (CRP) is independently associated with the occurrence of atrial fibrillation [3], and elevated tumor necrosis factor-α (TNF-α) and interleukin-6 (IL-6) also increase the risk of type 2 diabetes mellitus, coronary heart disease and atherosclerosis [4]. In addition, they are important motor response factors, and they play an important mediating role in the improvement of CVD by exercise.

Circadian rhythms are closely related to cardiovascular metabolism [5], the occurrence, treatment, and prognosis of CVD are all regulated by the biological clock [6], and circadian disturbances such as shift work and insomnia may exert adverse effects on cardiovascular metabolisms, such as disorders of the TG metabolism, dysglycemia, and elevated levels of biomarker resistin in atherosclerosis [7]. At the molecular level, blood pressure, blood lipid metabolism, inflammatory cytokines, and cardiac autonomic nerves are all regulated by circadian clock genes [8] so their peak and trough levels fluctuate regularly over 24 h periods, which leads to differences in the frequency of cardiovascular events during the morning, noon, and evening. Endogenous circadian rhythm disturbance may manifest in extrinsic cardiovascular events; previous studies have found that the risk of developing hypertension, etc., is higher during the early morning, while mortality in patients with heart disease peaks during the afternoon [9]. Thus, based on the characteristics of the diurnal fluctuations in the expression of cardiovascular metabolic markers and the circadian rhythms in cardiovascular risk, the proper organization of an individual’s daily activities (such as work, diet, and physical exercise) may help improve their cardiovascular health.

Physical activity as an effective means to prevent and treat cardiovascular and metabolic diseases has been widely recognized worldwide. Its mechanism involves endocrine, metabolism, immune and other systems [10]. Most previous clinical experimental studies have found that both regular long-term physical activity and acute high-intensity training could increase the plasma high-density lipoprotein cholesterol (HDL-C) levels, reduce the postprandial triglyceride (TC) levels, regulate the blood sugar levels, and improve the body’s inflammatory response in CVD patients [11,12]. Cross-sectional and cohort studies in various countries have also confirmed that moderate exercise can reduce the risk of coronary heart disease, heart failure, obesity, and T2DM, etc., thereby reducing mortality and improving the prognosis of relevant populations [13,14,15,16]. To maximize the improvement of cardiovascular metabolic risk markers levels, relevant guidelines and an expert consensus have given specific recommendations in terms of the mode, cycle, intensity, and duration of exercise training. However, the timing of exercise is often an overlooked factor, and the guidelines also rarely mention when exercise can minimize the cardiovascular risk factors [17,18]. The effects of physical activity on cardiovascular metabolic health may depend on the circadian rhythm. It is known that exercise is a powerful zeitgeber that mediates the circadian clock in mammals, which can induce the expression of the circadian genes in organs and tissues and change the phase of melatonin and visceral temperature, thus affecting a series of physiological processes in the body. As the expression of the cardiovascular risk factors show a 24 h fluctuation, therefore, exercise at different times of the day may also have different effects on cardiovascular metabolic health. Sato et al. [19] used transcriptomic and metabolomic methods and revealed the specific circadian effects of exercise-induced glycolysis and lipid metabolism reactions and their related effects between the organs of mice, and different responses of the cardiac metabolic risk markers to morning exercise (ME) and evening exercise (EE) were also found in human studies [20,21], suggesting the potential of timed exercise in reducing the cardiovascular risk factors. Based on this, this article provides a systematic review on how exercise at different times of day affects the cardiovascular risk factors in adults, and it aims to further clarify the role of exercise as a “time therapeutic agent” to provide a reference for the formulation of finer exercise prescriptions for subsequent relevant groups.

## 2. Method

The current review conforms to the Preferred Reporting Items for Systematic Reviews and Meta-Analyses (PRISMA) guidelines [22].

### 2.1. Search Strategy

The systematic review of the literature was conducted using PubMed, Web of Science, Embase, and CNKI databases. Taking the example of Web of Science, the following search syntax was used to identify potential articles: (“exercise timing” OR “timed exercise” OR “diurnal time of exercise” OR “exercise at different times of the day” OR “morning exercise” OR “afternoon exercise” OR “evening exercise” OR “morning versus afternoon exercise” OR “morning versus evening exercise”) AND (“waist circumference” OR “total cholesterol” OR “triglyceride” OR “low-density lipoproteins cholesterol” OR “high-density lipoproteins cholesterol” OR “blood lipids” OR “blood pressure” OR “blood sugar” OR “C-reactive protein” OR “interleukin-6” OR “inflammatory factor” OR “cardiovascular risk factors”). The search included articles that were published between January 2000 and June 2022. In addition, we screened the reference of included articles for relevant or potentially eligible studies.

### 2.2. Inclusion and Exclusion Criteria

Two reviewers independently reviewed the articles that were identified using the search and extracted the articles that met the inclusion criteria; the study was excluded if it was not written in English or if it was not published in a peer-reviewed journal. According to the PICOS framework, the literature inclusion and exclusion criteria for this systematic review are as follows.

Population: There were no restrictions on disease, and all adult populations (≥18 years old) met the inclusion criteria for this review, except for those with congenital immunodeficiency.

Intervention: The interventions that were included in this review should have had an independent study of the exercise training component with or without human guidance or the intervention of caloric restriction or medication. There were no restrictions on specific exercise methods (aerobic training, resistance training, etc.), exercise intensity, and frequency, while the circadian time point of the exercise intervention should have been specifically reported.

Control: The included studies needed to have control groups and they were only used if they could be compared.

Outcomes: The studies that were included in this review needed to have clear pre-and post-intervention measures. The outcomes mainly included: low-density lipoproteins cholesterol (LDL-C), high-density lipoproteins cholesterol (HDL-C), triglyceride (TG), total cholesterol (TC), blood glucose (BG), systolic pressure (SBP), diastolic pressure (DBP), C-reactive protein (CRP), tumor necrosis factor-α (TNF-α), interleukin-6 (IL-6), and waist circumference (WC). These outcome measures are currently widely studied mediators of exercise in promoting cardiovascular health.

Study type: The review included pilot studies, both prospective randomized and non-randomized controlled studies, and excluded single-arm experiment, cross-sectional, and retrospective studies.

### 2.3. Literature Selection and Submission of Materials

Two researchers independently screened the included literature and extracted data, if there is any disagreement, it shall be resolved through discussion or consultation with a third party. The extracted content mainly includes the year of the included study, authors, sample characteristics, interventions in the experimental and comparator group (time of day, exercise mode, period, intensity, and frequency), measurement outcomes of cardiovascular risk factors, and adverse events.

### 2.4. Risk Assessment of Literature Bias

Two researchers independently completed the literature risk assessment of the bias, and if a consensus could not be reached, the decision was made by another senior researcher. The risk of bias in the included randomized controlled studies were assessed by using the Cochrane bias risk assessment tool [23], which was mainly assessed from the aspect of random sequence generation, allocation concealment, the blinding of researchers and subjects, the blinding of outcome testing, complete outcome data, and other biases, and the results were rated as “low risk”, “high risk”, or “unclear risk”.

The risk of bias for non-randomized controlled studies were assessed by using the ROBIN-S bias risk assessment scale [24], which was mainly judged from the aspect of confounding bias, selection bias, a bias in the measurement classification of interventions, a bias due to deviations from the intended interventions, a bias due to missing data, a bias in the measurement of outcomes, and a bias in the selection of the reported result, and the results were rated as “low risk”, “moderate risk”, “serious risk”, and “critical risk”.

## 3. Result

### 3.1. Literature Search

The search strategy initially yielded 1038 articles from four databases, of which 789 duplicates were discarded. Through the independent screening of the abstracts by two authors, 58 studies were considered relevant to the topic. Then, they read the full text; 42 studies were excluded for the following reasons: they had the wrong outcomes (*n* = 28), they included no controlled trials (*n* = 3), they were not written in English (*n* = 2), they were of an inappropriate article type (*n* = 3), they included congenital immunodeficiency (*n* = 1), and they were of an inappropriate article type (*n* = 33). Sixteen studies met the inclusion criteria and were finally included in this review. The inter-rater agreement was 93.75% after the first screening, and this reached 100% consensus following the common discussion. The flow chart of the literature retrieval and screening process is presented in Figure 1.

### 3.2. Study Characteristics

The characteristics of the literature are shown in Table 1. Of these, 11 trials adopted randomized controlled trial (RCT) designs, while the rest were non-RCT designs. The sample size ranged from 11–275 participants who were diagnosed with obesity, hypertension, T2DM, and CAD, while the subjects in four of the studies did not report any metabolic or cardiovascular disease. Four studies conducted trials of acute exercise interventions, while the remaining intervention periods ranged from 12 days to 12 weeks. The exercise interventions included aerobic training, resistance training, aerobic training, which was combined with resistance training, compound exercise, and high-intensity interval exercise (HIIE). The training frequency varied from 2–5 times/week, and the exercise intensity ranged from moderate to a high intensity which was mainly measured by using metrics such as VO_2peak_, HR_max_, and 1RM. Specific intervention diurnal time points were reported in all of the included trials, except for Matthew et al. wherein the use of “before and after breakfast and dinner” was a time classification basis.

### 3.3. Quality Assessment

The risk of literature bias is shown in Figure 2 and Table 2. The 11 included RCTSs all have a certain risk of bias. Since it is difficult to achieve participant blinding in exercise intervention experiments, we gave this term a “high risk” unless it is otherwise stated in the text. The selection bias, attrition bias, and self-report bias were of low risk in all of the include studies, while four studies reported allocation concealment, and four studies reported blinded outcome measurers. There were potential additional risks of bias in three of the studies, such as the lack of effective monitoring of the subjects’ compliance with training.


ijerph-19-14194-t001_Table 1Table 1Time-dependent effects of physical activity on cardiovascular risk factors in adults.Study DetailsParticipantsChronotype AssessmentTime of ExerciseExercise InterventionOutcomesKey FindingsArciero [25], 2022, USA, RCT56 Healthy adultsnot assessedME: 6:30–8:30 a.m.EE: 6:00–8:00 p.m.RISE60 min/session4 times/weekfor 12 weeksSP (mmHg)DP (mmHg)IL-6 (pg/mL)CRP (µU/mL)glucose (mg/dL)ME vs. EE (women): −12.5 ± 2.7 vs. 2.3 ± 3 (*p* < 0.05)ME vs. EE (men): −3.5 ± 2.6 vs. −14.9 ± 5.1 (*p* < 0.05)ME vs. EE (women): −10 ± 1 vs. −5 ± 5 (*p* < 0.05)ME vs. EE (men): No significant between-group differences (*p* > 0.05)No significant changes in both groups (*p* > 0.05)Kim [35], 2022, Japan, Non-RCT14 healthy men(21.8 ± 0.2 years)MEQME: 9:00–11:00 a.m.AE: 4:00–6:00 p.m.Aerobic exercise(treadmill)60% VO_2_ peakfor 65 minTG (mg/dL)HDL-C (mg/dL)LDL-C (mg/dL)Glucose (mg/dL)ME (post) vs. ME (pre): 123.9 ± 15.1 vs. 129.8 ± 7.9 (*p* > 0.05)AE (post) vs. AE (pre): 96.2 ± 13.5 vs. 129.8 ± 7.9 (*p* < 0.05)ME (post) vs. ME (pre): 60.0 ± 3.8 vs. 53.9 ± 3.1 (*p* > 0.05)AE (post) vs. AE (pre): 60.9 ± 4.9 vs. 53.9 ± 3.1 (*p* < 0.05)ME (post) vs. ME (pre): 87.4 ± 6.8 vs. 89.9 ± 6.7 (*p* > 0.05)AE (post) vs. AE (pre): 86.1 ± 6.2 vs. 89.9 ± 6.7 (*p* > 0.05)ME (post) vs. ME (pre): 86.1 ± 6.1 vs. 88.5 ± 5.0 (*p* > 0.05)AE (post) vs. AE (pre): 96.1 ± 6.0 vs. 88.5 ± 5.0 (*p* > 0.05)Teo [26], 2021, Austria, RCT40 obese adults (51 ± 13 years)not assessedME: 8:00–10:00 a.m.EE: 5:00–7:00 p.m.Aerobic exercise (treadmill) combined with resistance exercise60 min/session3 times/weekfor 12 weeksWC (cm)ME (post) vs. ME (pre): 97.6 ± 9.1 vs. 101.0 ± 9.6 (*p* < 0.05)EE (post) vs. EE (pre): 99.5 ± 11.6 vs. 103.8 ± 11.2 (*p* < 0.05)ME vs. EE: −3.5 ± 1.8 vs. −4.3 ± 2.8 (*p* > 0.05)Saidi [27], 2021, France, RCT28 overweight adults (ME: 54.7 ± 8.5 years; 53.7 ± 7.5years)MEQME: 9:00 a.m.EE: 6:30 p.m.Aerobic exercise (elliptical bike, rower, treadmill) combined with resistance exercise 90 min/session3 times/weekfor 12 weeksWC (cm)ME (post) vs. ME (pre): 117.1 ± 13.3 vs. 120.5 ± 14.3 (*p* = 0.03)EE (post) vs. EE (pre): 114.2 ± 12.0 vs. 115.9 ± 12.1 (*p* = 0.11)Matthew [36], 2020, Canada, Non-RCT14 individualswith T2DM (65 ± 9 years)not assessedME: Before breakfast AE: Before dinnerEE: After dinnerAerobic exercise (walking)5.0 km/h50 minfor 12 daysFasting glucose (mmol/L)Mean 24 h glucose (mmol/L)AE vs. ME: 7.3 ± 1.2 vs. 6.8 ± 1.0 (*p* = 0.003)AE vs. EE: 7.3 ± 1.2 vs. 7.1 ± 1.2 (*p* = 0.404)ME vs. EE: 6.8 ± 1.0 vs. 7.1 ± 1.2 (*p* = 0.979)ME/AE/EE vs. CON: 6.8 ± 1.0/7.3 ± 1.2/7.1 ± 1.2 vs. 6.9 ± 1.2(*p* = 0.388, *p* = 0.223, *p* = 0.522)AE vs. ME: 7.3 ± 0.9 vs. 7.5 ± 0.7 (*p* = 0.570)AE vs. EE: 7.3 ± 0.9 vs. 7.6 ± 0.8 (*p* = 0.423)ME vs. EE: 7.5 ± 0.7 vs. 7.6 ± 0.8 (*p* = 0.824)ME/AE/EE vs. CON: 7.5 ± 0.7/7.3 ± 0.9/7.6 ± 0.8 vs. 7.5 ± 0.6(*p* = 0.347, *p* = 0.01, *p* = 0.447)Irandoust [28], 2020, Turkey, RCT15 obese women (46.9 ± 5.2 years)not assessedME: 7:00–9:00 a.m.EE: 6:00–8:00 p.m.Aerobic exercise(jogging)30 min/session3 times/weekfor 2 weeksWC (cm)ME vs. EE: −1.13 ± 2.44 vs. 0.03 ± 3.14 (*p* = 0.345)Teo [29], 2019, Austria, RCT40 overweight adults with or without T2DM (51 ± 13 years)not assessedME: 8:00–10:00 a.m.EE: 5:00–7:00 p.m.Aerobic exercise (treadmill walking) combined with resistance exercise60 min/session3 times/weekfor 12 weeksFasting glucose (mmol/L)ME (post) vs. ME (pre): 6.78 ± 1.45 vs. 7.68 ± 1.70 (*p* < 0.05)EE (post) vs. EE (pre): 7.10 ± 2.41 vs. 8.28 ± 3.72 (*p* < 0.05)ME vs. EE (overall): −0.27 ± 0.24 vs. 0.25 ± 0.23 (*p* > 0.05)ME vs. EE (T2DM): −1.27 ± 0.75 vs. −1.80 ± 1.77 (*p* > 0.05)Chiang [30], 2019, Taiwan, RCT20 individualswith T2DM (48.5 ± 4.2 years)not assessedME: 8:00–10:00 a.m.AE: 2:00–4:00 p.m.EE: 6:00–8:00 p.m.Aerobic exercise(treadmill)70% HRmax40–50 min/session3 times/weekfor 12 weeksEIGR (mg/dL)ME vs. AE: 73.9 ± 28.2 vs. 55.5 ± 31.3(*p* < 0.05)AE vs. EE: 55.5 ± 31.3 vs. 75.0 ± 38.7(*p* > 0.05)ME vs. EE: 73.9 ± 28.2 vs. 75.0 ± 38.7(*p* > 0.05)Savikj [37], 2018, Sweden, Non-RCT11 individualswith T2DM (65 ± 9 years)not assessedME: 8:00 a.m.EE: 4:00 p.m.HIIE (cycling)7 min warm-up + 6 pulses (above 220 W, range 180–350 W)4 times/weekfor 2 weeksLDL-c (mmol/L)HDL-c (mmol/L)TG (mmol/L)TC (mmol/L)Glucose (mmol/L)ME vs. AE vs. Pre: 2.4 ± 0.4 vs. 2.3 ± 0.4 vs. 2.4 ± 0.4 (*p* > 0.05)ME vs. AE vs. Pre: 1.3 ± 0.1 vs. 1.2 ± 0.1 vs. 1.2 ± 0.1 (*p* > 0.05)ME vs. AE vs. Pre: 1.6 ± 0.3 vs. 1.4 ± 0.2 vs. 1.2 ± 0.2 (*p* > 0.05)ME vs. AE vs. Pre: 4.4 ± 0.3 vs. 4.2 ± 0.4 vs. 4.2 ± 0.4 (*p* > 0.05)ME vs. AE vs. Pre: 7.7 ± 0.4 vs. 7.5 ± 0.3 vs. 7.3 ± 0.3 (*p* > 0.05)Brito [38], 2018, Brazil, RCT50 treatedhypertensive men(ME: 51 ± 8 yearsEE: 49 ± 8 yearsCON: 50 ± 9 years)Horne and Ostberg questionnaireME: 7:00–9:00 a.m.EE: 6:00–8:00 p.m.Aerobic exercise(cycling)Moderate intensity45 min/session3 times/weekfor 10 weeksSBP (mmHg)DBP (mmHg)24 h BP (mmHg)Asleep DBP (mmHg)EE vs. ME vs. Con (MA): −5.0 ± 6.0 vs. ME vs. Con (*p* < 0.05)EE vs. ME vs. Con (EA): −8.0 ± 7.0 vs. ME vs. Con (*p* < 0.05)EE vs. ME vs. Con: no significant differences (*p* > 0.05)EE vs. ME vs. Con: −3.0 ± 5.0 vs. ME vs. Con (*p* < 0.05)EE vs. ME vs. Con: −3.0 ± 4.0 vs. ME vs. Con (*p* < 0.05)Bohumila [31], 2018, Slovakia, RCT31 healthy olderWomen(66 ± 4 years)not assessedME: 7:30 a.m.EE: 6:00 p.m.progressive strengthtraining2 times/weekfor 12 weeksLDL-c (mmol/L)HDL-c (mmol/L)TG (mmol/L)CRP (mmol/L)Glucose (mmol/L)ME (post) vs. ME (pre): 3.0 ± 0.9 vs. 2.6 ± 0.8 (*p* > 0.05)EE (post) vs. EE (pre): 3.6 ± 1.4 vs. 3.4 ± 1.3 (*p* > 0.05)ME (post) vs. ME (pre): 1.5 ± 0.4 vs. 1.5 ± 0.5 (*p* > 0.05)EE (post) vs. EE (pre): 1.7 ± 1.4 vs. 1.5 ± 0.4 (*p* > 0.05)ME (post) vs. ME (pre): 1.6 ± 0.4 vs. 1.4 ± 0.5 (*p* < 0.01)EE (post) vs. EE (pre): 1.3 ± 0.7 vs. 1.7 ± 1.0 (*p* < 0.01)ME (post) vs. ME (pre): 1.8 ± 1.1 vs. 2.0 ± 1.2 (*p* > 0.05)EE (post) vs. EE (pre): 2.0 ± 1.4 vs. 2.5 ± 1.8 (*p* > 0.05)ME (post) vs. ME (pre): 5.6 ± 0.6 vs. 5.8 ± 0.4 (*p* < 0.05)EE (post) vs. EE (pre): 5.1 ± 0.5 vs. 5.6 ± 0.9 (*p* < 0.05)Alizadeh [32], 2017, Iran, RCT48 overweight females (ME: 33.56 ± 5.98 years; EE: 33.89 ± 6.58 years)not assessedME: 8:00–10:00 a.m.AE: 2:00–4:00 p.m.Aerobic exercise(treadmill)VT heart rate30 min/session3 times/weekfor 6 weeksWC (cm)ME (post) vs. ME (pre): 83.79 ± 1.13 vs. 85.78 ± 0.98 (*p* < 0.05)EE (post) vs. EE (pre): 85.08 ± 1.37 vs. 86.30 ± 1.30 (*p* > 0.05)Brito [33], 2015, Brazil, Non-RCT16 pre-hypertensive men(32 ± 7 years)Horne and Ostberg questionnaireME: 7:30–11:30 a.m.EE: 5:00–9:00 p.m.Aerobic exercise(cycling)50% VO2 peakfor 45 minSBP (mmHg)DBP (mmHg)ME (post) vs. ME (pre): 118 ± 3 vs. 122 ± 9 (*p* < 0.05)ME (post) vs. ME (con): 118 ± 3 vs. 124 ± 9 (*p* < 0.05)EE (post) vs. EE (pre): 117 ± 4 vs. 123 ± 9 (*p* < 0.05)EE (post) vs. EE (con): 117 ± 4 vs. 119 ± 9 (*p* < 0.05)ME vs. EE: −7 ± 3 vs.−3 ± 4 (*p* < 0.05)ME (post) vs. ME (pre): 84 ± 8 vs. 84 ± 8 (*p* > 0.05)ME (post) vs. ME (con): 84 ± 8 vs. 87 ± 6 (*p* < 0.05)EE (post) vs. EE (pre): 83 ± 8 vs. 84 ± 6 (*p* > 0.05)EE (post) vs. EE (con): 83 ± 8 vs. 86 ± 7 (*p* < 0.05)ME vs. EE: −3 ± 3 vs. −3 ± 3 (*p* > 0.05)Kim [39], 2015, Japan, Non-RCT14 healthy men(24.3 ± 0.2 years)not assessedME: 9:00–10:00 a.m.EE: 5:00–6:00 p.m.Aerobic exercise(treadmill)60% VO2 peakfor 65 minIL-6 (pg/mL)CPR (mg/L)TNF-α (mg/L)significantly improved in EE compared to ME (*p* < 0.05)ME (post 2 h) vs. ME (post) vs. ME (pre): 0.60 ± 0.12 vs. 0.56 ± 0.13 vs. 0.59 ± 0.14 (*p* > 0.05)EE (post 2 h) vs. EE (post) vs. EE (pre): 0.62 ± 0.14 vs. 0.60 ± 0.12 vs. 0.52 ± 0.12 (*p* > 0.05)ME (post 2 h) vs. ME (post) vs. ME (pre): 0.04 ± 0.01 vs. 0.04 ± 0.01 vs. 0.04 ± 0.01(*p* > 0.05)EE (post 2 h) vs. EE (post) vs. EE (pre): 0.05 ± 0.01 vs. 0.05± 0.01 vs. 0.05 ± 0.01 (*p* > 0.05)ME (post 2 h) vs. ME (post) vs. ME (pre): 0.04 ± 0.01 vs. 0.04 ± 0.01 vs. 0.04 ± 0.01 (*p* > 0.05)EE (post 2 h) vs. EE (post) vs. EE (pre): 0.05 ± 0.01 vs. 0.05 ± 0.01 (*p* > 0.05)Lian [34], 2014, China, RCT275 individualswith CAD (ME: 64 ± 9 years; EE: 62 ± 10 years; CON: 50 ± 9 years)not assessedME/AE/EEAerobic exercise(walking)Modeatre intensity30 min/session5 times/weekfor 12 weeksLDL-c (mmol/L)HDL-c (mmol/L)TG (mmol/L)TC (mmol/L)Glucose (mmol/L)ME (post) vs. ME (pre): 0.63 ± 0.22 vs. 0.92 ± 0.29 (*p* < 0.001)EE (post) vs. EE (pre): 0.55 ± 0.19 vs. 0.90 ± 0.32 (*p* < 0.001)ME (post) vs. EE (post) vs. CON: 0.63 ± 0.22 vs. 0.55 ± 0.19 vs. 0.79 ± 0.19 (*p* < 0.001)ME (post) vs. ME (pre): 0.12 ± 0.24 vs. 0.06 ± 0.22 (*p* = 0.022)EE (post) vs. EE (pre): 0.14 ± 0.23 vs. 0.07 ± 0.23 (*p* < 0.001)ME (post) vs. EE (post) vs. CON:0.12 ± 0.24 vs. 0.14 ± 0.23 vs. 0.14 ± 0.25 (*p* = 0.824)ME (post) vs. ME (pre): 0.15 ± 0.42 vs. 0.32 ± 0.38 (*p* < 0.001)EE (post) vs. EE (pre): 0.22 ± 0.39 vs. 0.35 ± 0.42 (*p* < 0.001)ME (post) vs. EE (post) vs. CON: 0.15 ± 0.42 vs. 0.22 ± 0.39 vs. 0.32 ± 0.39 (*p* = 0.018)ME (post) vs. ME (pre): 1.20 ± 0.17 vs. 1.43 ± 0.24 (*p* < 0.001)EE (post) vs. EE (pre): 1.17 ± 0.15 vs. 1.42 ± 0.27 (*p* < 0.001)ME (post) vs. EE (post) vs. CON: 1.20 ± 0.17 vs. 1.17 ± 0.15 vs. 1.29 ± 0.13 (*p* < 0.001)ME (post) vs. ME (pre): 3.51 ± 1.11 vs. 2.77 ± 0.86 (*p* < 0.001)EE (post) vs. EE (pre): 3.52 ± 0.89 vs. 2.36 ± 0.79 (*p* < 0.001)ME(post) vs. EE(post): 0.74 ± 0.92 vs. 1.16 ± 1.07 (*p* = 0.038)EE(post) vs. CON: 1.16 ± 1.07 vs. 0.18 ± 0.95 (*p* < 0.001)David [40], 2011, UK, Non-RCT14 healthy adults (20.0 ± 1.6years)not assessedME: 8:15–9:00 a.m.EE: 6:15–7:00 p.m.Resistance exercise(3 sets of 8–12repetitions for 4 resistance exercises at 70% of 1 RM)for 30 minIL-6 (pg/mL)ME (post) vs. ME (pre): 4.42 ± 0.10 vs. 4.51 ± 0.18 (*p* > 0.05)ME (post) vs. ME (con): 4.63 ± 0.31 vs. 4.64 ± 0.44 (*p* > 0.05)RISE: resistance, interval, stretching, and endurance; MEQ: morningness-eveningness questionnaire; ME: morning exercise; EE: evening exercise; AE: afternoon exercise; VO_2peak_: maximum oxygen uptake; HR_max_: maximum heart rate; EIGR = exercise-induced blood glucose response.
ijerph-19-14194-t002_Table 2Table 2Risk of bias assessment for non-randomized controlled trials.AuthorCounfounding BiasSelection BiasBias in Measurement Classification of InterventionsBias Due to Deviations from Intended InterventionsBias Due to Missing DataBias in Measurement of OutcomesBias in Selection of the Reported ResultKim 2022 [35]low risklow risklow risklow risklow riskmoderate risklow riskMatthew 2020[36] moderate risklow risklow riskmoderate risklow riskmoderate risklow riskSavikj 2018[37]moderate risklow risklow risklow risklow riskmoderate risklow riskBrito 2015[38]low risklow risklow risklow risklow riskmoderate risklow riskKim 2015[39]moderate risklow risklow risklow risklow riskmoderate risklow riskDavid 2011[40]moderate risklow risklow risklow risklow riskmoderate risklow risk


Of the six included self-controlled experiments, since none of them reported whether the outcome measurers were blinded, we assigned a moderate risk label to this dimension. The selection bias, a bias in the classification of interventions, and a bias due to missing data were of low risk in all of the included studies. One study had a low risk of bias due to there being deviations from the intended interventions; four studies showed no potential confounding bias.

## 4. Outcome

### 4.1. Blood Pressure

Three studies reported a blood pressure (BP)-related outcome. Arciero et al. [25] conducted a 12-week compound training intervention in healthy subjects and observed that significant between-group differences existed for the SBP (−3.5 ± 2.6 vs. −14.9 ± 5.1, mmHg) in men and SBP (−12.5 ± 2.7 vs. 2.3 ± 3, mmHg) and the DBP (−10 ± 1 vs. −5 ± 5, mmHg) in women, ME vs. EE, respectively. In a 10-week cycling exercise intervention trial in individuals with hypertension, only the subjects who were assigned to the EE group showed significant reductions in the SBP (−5.0 ± 0.6,−8.0 ± 0.6, mmHg), 24 h BP (−3.0 ± 0.5, mmHg), and asleep BP (−3.0 ± 4.0, mmHg) when they were compared with EE and Con group [38]. Brito et al. [33] conducted a moderate-intensity acute aerobic exercise intervention in 16 pre-hypertensive men and found that training at any time can produce hypotensive net effects, but the SP showed a greater decrease in the ME group (−7 ± 3 vs. −3 ± 4, mmHg).

### 4.2. Blood Lipids

Four studies comprising 331 subjects examined the effect of ME vs. AE/EE on blood lipids in adults. A non-RCT showed that significant interactions existed for TG (96.2 ± 13.5 vs. 129.8 ± 7.9, mg/dL) and the HDL-C (60.9 ± 4.9 vs. 53.9± 3.1, mg/dL) indexes in the AE group, while there were no significant differences in the other indexes of lipid metabolism between and within the groups [35]. Bohumila et al. [31] conducted a 12-week resistance training and found that ME significantly increased TG (1.6 ± 0.4 vs. 1.4 ± 0.5, mmol/L) in healthy older women, whereas EE reduced it (1.3 ± 0.7 vs. 1.7 ± 1.0, mmol/L). A separate 12-week walking program lead to a significant improvement in the TG, TC, HDL-C, LDL-C indexes from their baselines in individuals with CAD, and the EE group showed a greater favorable change in their LDL-c [34]. It is worth noting that Mladen et al. [37] observed that neither the morning nor evening HIIT was effective in improving the blood lipids in the subjects.

### 4.3. Blood Glucose

In terms of blood glucose, four of the studies did not report significant changes from the baseline [31,34,35,37]. Among the rest studies, Matthew et al. [36] divided healthy older women into ME, EE, and CON groups, and the intervention groups accepted resistance training for 12 weeks; the results of this study showed that both the ME and EE groups significantly improved the blood glucose levels of the subjects, and the decrease was more pronounced in the EE group (−4 ± 6% vs. −8 ± 10%, mmol/L). In another long-term aerobic training experiment for individuals with T2DM [30], it was found that exercise training in the afternoon significantly increased the EIGR level of the subjects when they were compared with those in the ME group (*p* = 0.004).

### 4.4. Waist Circumference

Four studies comprising 131 subjects examined the effect of ME/EE on the WC in individuals with obesity. A 6-week aerobic exercise intervention study observed that ME (83.79 ± 1.13 vs. 85.78 ± 0.98, cm) was more effective than EE (85.08 ± 1.37 vs. 86.30 ± 1.30, cm) was in reducing the WC of the overweight subjects [32]; the same finding was reported by Saidi et al. (117.1 ± 13.3 vs. 120.5 ± 14.3, cm) [27]. Following a 12-week aerobic training session that was combined with a resistance training session, Teo et al. [26] and Irondoust et al. [28] found favorable changes in reducing the WC in both the ME and EE groups, but no significant differences were observed between the groups (−3.5 ± 1.8 vs. −4.3 ± 2.8, −1.13 ± 2.44 vs. 0.03 ± 3.14, cm, respectively).

### 4.5. Inflammatory Cytokines

Among the six studies measuring the inflammatory factors-related outcomes, four of them did not report a significant reduction in the inflammatory cytokines in the exercise group [25,31,40]. Kim et al. [39] performed an acute, moderate intensity aerobic exercise intervention in 14 healthy people and observed that there were no significant changes in the TNF-α and CRP in the exercise group when they were compared with baseline, while the IL-6 significantly improved, and the increase in the EE group was more pronounced than it was in the ME group. Similarly, in the study by Lian et al. [34], long-term walking in the evening resulted a larger reduction in the CRP levels in the CAD patients than when they were compared with those that performed it in the early morning (1.16 ± 1.07 vs. 0.18 ± 0.95, mg/L.

## 5. Discussion

As far as we know, this is the first systematic review to provide a compassion symmetry of how physical activity at different circadian times of day modulates the levels of the cardiovascular risk biomarkers. Considering the limited amount of literature for a specific population, the subjects that were included in this study covered adults, regardless of whether they had metabolic and cardiovascular diseases.

### 5.1. Time-Dependent Effects of Physical Activity on Cardiovascular Risk

#### 5.1.1. Blood Glucose

Our review indicates that blood glucose levels may be affected by the exercise timing as two studies supported that AE/EE was more favorable for glycemic control than ME was [30,36]. Physical exercise is known to help reduce the body fat content, which in turn regulates insulin sensitivity and increases the muscle glucose utilization, thereby lowering the blood sugar levels [41,42,43]. However, in the strength training trial of healthy people, Bohumila et al. [31] found similar levels of body fat loss in the subjects in the ME and EE groups, suggesting that other factors may mediate this difference, such as hormones and energy utilization mode. Previous studies from the metabolomics field found that exercise in mice in the early active phase significantly increases glycolysis when it was compared to the early resting phase, while increasing the utilization of carbohydrates and ketone bodies since the circadian rhythms of humans and rodents are opposite to each other, it can be inferred from the phase difference that physical exercise in the evening is more conducive to controlling blood sugar [19]. However, it should be noted that due to the disorder of the circadian rhythm of glucose metabolism in people with T2DM, liver glucose is abnormally elevated in the morning, resulting in an increase in blood sugar in the morning [44], while blood sugar in normal people is higher in the evening. Nonetheless, both Bohumila et al. [31] and Chiang et al. [30] found that AE/EE was more helpful for blood sugar control in different subjects, which may be caused by the differences in the muscle use patterns and molecular signaling between the two training methods. Four studies did not report any significant changes from the baseline [31,34,35,37] due to the large heterogeneity of the exercise intensity, period, and the time interval between the exercise and the previous meal between the studies, therefore, it is difficult to explain the difference between the results. In addition, the authors suggested that a standardized diet may result in a reduced energy intake, which in turn led to lower blood sugar levels in the subjects.

#### 5.1.2. Blood Lipids

Long-term regular exercise can positively regulate blood lipid metabolism, reduce the concentration of TG in the blood, and increase the level of HDL-C; the mechanism of this is mainly related to the increased lipoprotein lipase activity which is responsible for hydrolyzing chylomicrons and VLDL TAG in granules [45,46]. The results from the studies that we included in this review are consistent with previous scholars’ findings [47,48], except for Mladen et al. [37] who did not observe significant changes in any of the subjects’ lipid metabolism indicators in their 2-week HIIE session, which may be due to the short intervention period, however, previous studies found that although short-term exercise can immediately reduce the blood lipids, the effect has not been continuously superimposed, and the chronic cumulative effect of energy consumption may require a long exercise cycle for it to appear. Two studies supported that EE was more beneficial for lipid metabolism [31,34], additionally, studies of mice models revealed the different effects of aerobic exercise at different times of the day on the lipid metabolism in the liver, serum, IWAT, and EWAT when they were compared to ME, while EE stimulated the rise of acylcarnitine levels in skeletal muscle more effectively, thus increasing the muscle demand for energy sources other than glycolysis [19]. It is worth noting that Bohumila et al. found an increase in TG from the baseline in healthy subjects in the ME group, but the result is difficult to interpret based on existing measurements; the increase in TG following the resistance training may be due to the increased concentrations of inflammatory factors such as IL-6 and TNF-α resulting from a muscle injury [49], which can reduce the rate of fat mobilization which is inhibited by insulin, and the possibility that this can happen during the early morning rather than during the late evening may be due to the higher viscosity of the skeletal muscle in the early morning after a full night’s rest, but this needs to be tested further.

#### 5.1.3. Blood Pressure

Physical exercise is an important non-drug prescription for the prevention and treatment of hypertension, which may be related to its ability to modulate angiotensin, enhance the arterial baroreceptor activity, and promote a sympathetic and vagal balance [50,51,52]. The results of our review are consistent with previous studies, both long-term regular exercise and short-term acute moderate-intensity aerobic exercise reduced the SBP and DBP to varying degrees. In a long-term intervention study by Brito et al. [33], morning exercise was observed to be more effective in reducing the SBP, which may be because ME mediated more cardiac sympathovagal homeostasis, thus resulting in a decrease in the cardiac output as the net effect of an increased systemic vascular resistance and improvements in the other heart rate variability markers were more pronounced after ME when it was compared with EE. Another 10-week cycling training trial that was conducted by the same team showed that EE had more potential to reduce the SBP, the 24 h BP and asleep BP in treated hypertensive men [38] owing to the large differences in the training cycles, while the sympathetic nervous system adaptation changes in vascular resistance and vasoconstriction in the latter group may not be as pronounced as the immediate changes in the former group are, which may lead to different results. In addition, the subjects in the latter experiment were more likely to choose to take anti-hypertensive drugs in the morning, thereby meaning that the post-exercise hypotension in the ME group might be influenced by the efficacy of the drug, furthermore, it may also be hidden by a circadian rise in the BP in the morning. Finally, gender is also a factor that cannot be ignored as reported in the trial of Arciero et al. [25].

#### 5.1.4. Waist Circumference

Waist circumference is not only closely related to central obesity, but also an important predictor of cardiovascular disease. The four long-term intervention studies that were included in this review indicated that regular physical activity can positively modulate the body composition and reduce the WC, which is consistent with the findings of previous studies [53,54,55,56]. Two studies showed that ME resulted in a greater reduction in the WC [27,32], which may be caused by improvements in energy intake and appetite control, and Ironoust et al. [28] found that following 12-week jogging, the subjects in the ME group had significantly lower perceived hunger than those in the EE group did, and a greater decrease of the daily intake of carbohydrates and fats in the ME group was observed in the study by Alizadeh et al. Two studies [26,28] did not report significant differences between the groups, and the authors did not provide a further explanation for the finding; different responses of the glycolipid metabolism and energy utilization to ME and EE may help explain the results, but the corresponding evidence was lacking. In addition, the level and phase change of melatonin secretion may also be an important factor contributing to the different outcomes; ME helps stimulate more melatonin secretion and causes its circadian phase to advance [57], and previous studies have found that melatonin could effectively activate brown fat, which inhibits white fat accumulation and helps to reduce the amount of body fat. Therefore, ME may lead to greater reductions in the WC by increasing the production of melatonin [58], however, whether the phase of melatonin secretion (earlier or later) has an effect on WC remains to be explored.

#### 5.1.5. Inflammatory Cytokines

In terms of the inflammatory factors, an acute endurance exercise study found that ME significantly increased the plasma IL-6 concentrations more than EE did [39], and the authors suggest that this is due to there being a higher level of plasma epinephrine in the former than there is in the latter (as observed in the experiment). IL-6 is not only a typical exercise cytokine, but it is also involved in body lipid metabolism [59]. Previous studies showed that IL-6-deficient transgenic mice were prone to obesity when they were compared with wild-type mice, and following 18 days of treatment with IL-6, the body masses of the transgenic mice were reduced to a great extent, and so the time-dependent effect of IL-6 expression on exercise may help to explain the more favorable changes of EE on lipid metabolism and WC reduction that are reported in several studies in this review. Similarly, Lian et al. [34] found that long-term EE was more helpful than ME was in reducing the plasma CRP levels, which may be related to the circadian oscillations of the hormones and inflammatory factors themselves. In addition, several studies did not report significant changes in TNF-α, CRP, and IL-6 from the baseline in the exercise group [25,31,40], owing to the different study sample populations, as well as there being a greater heterogeneity in the exercise intensity, cycle, and mode, and these may all contribute to these results, for example, the low levels of the subjects’ inflammatory factors or the insufficient exercise intensity.

In conclusion, EE seems to be more effective in reducing the WC in adults, but whether the response of other cardiovascular risk factors to the physical activity time points is time-specific is still controversial, and they may be affected by exercise patterns, individual health status, and chronotype, etc. It should also be noted that in several studies, the changes or trends in the cardiovascular risk factors among the subjects who exercised at the same point were not completely synchronized, which also highlights the complexity of the human circadian rhythm system.

### 5.2. How Does It Work

The biological clock is the change of various behaviors and physiological functions of the body in a cycle that is close to 24 h [60,61]. The central circadian clock which is located in the suprachiasmatic nucleus of the hypothalamus is the pacemaker of the circadian rhythm, which is directly activated by sunlight and regulates the peripheral clocks in other organs and tissues (liver, muscle, and pancreas, etc.) through neurohumoral pathways, which enables the synchronization of the circadian rhythm throughout the body [62,63,64]. At the molecular level, the circadian clock is driven by an autoregulatory feedback gene expression network that is integrated by positive and negative regulatory transcription loops, mainly including core clock genes such as Clock, Bmall, Per, Cry, and Reverbα, etc., and the downstream clock control genes that are regulated by them [65]. The clock gene network can effectively regulate the body’s glucose and lipid metabolism and inflammatory response, etc., and maintain the 24 h circadian fluctuations [66,67,68,69,70,71].

Exercise, as a powerful zeitgeber [72,73,74], can induce the expression of clock genes in the skeletal muscle and other metabolic organs and tissues and lead to significant changes in the hormonal responses and tissue-specific transcription and metabolism. In rodent experiments, aerobic exercise was observed to up-regulate the protein expression of the Bmal1 and Clock genes in mice that were fed a high-fat diet and it induced the expression of rate-limiting enzymes in the catabolism of glucose metabolism [75], in addition, the effect of an acute resistance exercise intervention on the expression of Cry, Clock, and Bmal1 in the skeletal muscle was also found [76]. Since the hormone levels, sympathetic and parasympathetic nerve activity, energy metabolites, and inflammatory cytokines in the body are regulated by the clock genes and the rhythm of their expression has obvious circadian fluctuations, this suggests that the synchronizing of the timing of exercise with the rhythm of these circadian responses or the expression of the clock genes may amplify the cardiovascular health benefits of physical activity. In this regard, Dennis et al. observed different changes in the phase of the circadian clock (delayed or advanced) in mice that exercised at ZT5 (5 h after the lights were turned on) and ZT11 (11 h after the lights were turned on) [77], thereby indicating that exercise circadian time regulates the molecular clock, but its effect on the clock-regulated expression of the metabolic target proteins, and then, how it enhances/weakens the cardiovascular activity rhythm has rarely been reported. The energy metabolism, hormones, and nerve conduction signals that are generated by the body during different exercise timing may be fed back to the corresponding biological clock system using different paths so that the physiological activity process of different organs and tissues can adapt to the needs of ME/AE/EE, however, this still needs further experimental verification.

This systematic review comprehensively summarizes the current knowledge on the association between exercise timing and cardiovascular risk factors in adults, however, the limitations of this article should be noted. First of all, considering the limited number of studies on a certain population, we set the target population to all adults without a congenital immunodeficiency, which means that the physical health status, BMI, age, and gender of the included sample among the different studies were relatively large heterogeneity, and the large differences in the exercise intervention methods, time, and intensity coupled with the small size for each of the outcomes resulted in no comparable samples, and so the results of the study should be interpreted with caution. In addition, only a few studies investigated morningness-eveningness in the subjects, while chronotype and exercise timing that is out of sync may increase the CVD risk such as high blood pressure, which may interfere with the interpretation of the experimental results.

## 6. Conclusions

The review found a piece of evidence that states that the cardiovascular health in adults may be time-dependent in response to physical activity, however, the findings are contradictory which may be due to us incurring several methodological issues or confounding factors. In addition, considering the differences in the cardiovascular molecular signaling among people with distinct health statuses and the complexity of the human circadian rhythm system, the findings should be interpreted with caution. In future research, under the premise of fully considering the chronotype and the characteristics of its circadian rhythm in a certain disease state, a large sample size and more rigorous RCT should be designed for a specific population to fully examine the effects of exercise on different time points in the day on cardiovascular health, thereby deepening our understanding of the temporal therapeutics of exercise.

## Figures and Tables

**Figure 1 ijerph-19-14194-f001:**
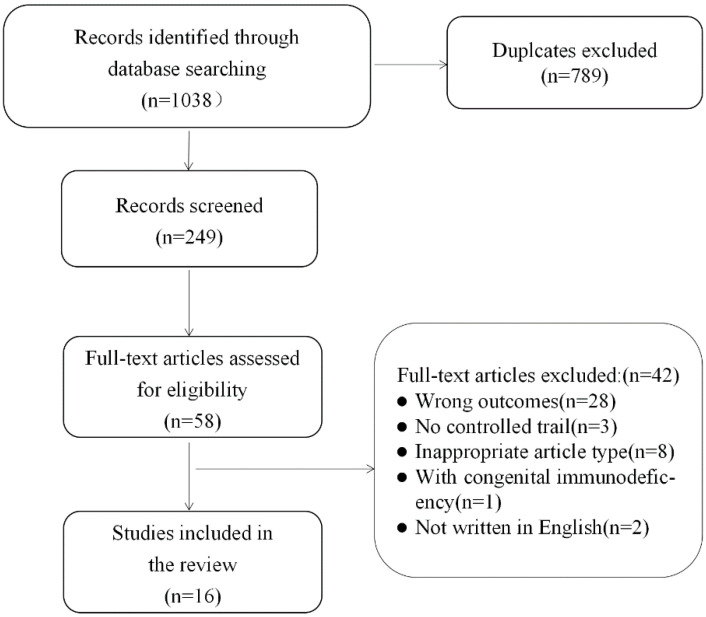
Literature search and screening process.

**Figure 2 ijerph-19-14194-f002:**
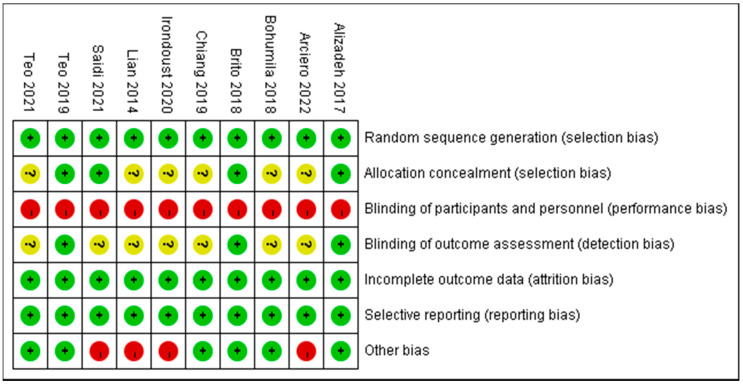
Risk of bias assessment for randomized controlled trials [25,26,27,28,29,30,31,32,33,34].

## Data Availability

Not applicable.

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
