# Peer review of "Time-Dependent Effects of Physical Activity on Cardiovascular Risk Factors in Adults: A Systematic Review"

_ijerph, 2022, doi:10.3390/ijerph192114194_

Round 1
Reviewer 1 Report
The objectives of this review are interesting and will be useful for exercise prescription. However, the introduction should be more explanatory (eg, justification for using only TNFalpha and IL-6 as markers of inflammation). Moreover, the relationship between exercise and work is also lacking, since the chronobiology of physical effort and work is well described and can contribute for this work.
The text in lines 59 to 62 is not understandable – it must be rewritten.
In the methodology, there must be a justification for moving from 1038 records identified to 249 works screened and then to only 58 works assessend for elegibility. The fact that the identified works had a reduced number of subjects should also be highlighted.
On page 15 there is an explanation for the influence of sunlight on the reduction of fat body or waist circumference. It seems like a far too forced explanation - Are people living in hot climates thinner?.
The work mentions the many biases and limitations, which is correct.
Author Response
Dear experts:
Thank you very much for your valuable comments on this article, the revision in strict accordance with the expert opinion,where the changes are made by the red font.Here are the details of the changes:
1) Following your request, We explain in the introduction section why the CRP,TNF-a and IL-6 were chosen as outcome indicators of inflammatory factors, mainly because they are the most extensive mediator cytokines studied in exercise-improved CVD, as shown in lines 41-43 and 46-48.
2) Following your request,we complement the relationship of exercise and biological rhythms, that exercise can regulate circadian rhythms, and that exercise at different times may have different effects on cardiovascular risk factors due to the rhythmicity of human physiological activities.
3)Following your request,We rewrote the original lines of 59-62 , as shown in lines 60-64,mainly highlight the importance of rationally arranging daily behavioral activities (such as exercise) according to the circadian characteristics of cardiovascular risk factors.
4)We described in detail the literature retrieval and exclusion process(repetitive literature, animal experiments, non-English texts, and so on) following you requested, as shown in lines 162-165.
5) Following your request,We rewrote the mechanism why morning exercise is more helpful to reducing body fat and waist circumference on page 15, probably because morning exercise is more conducive to the secretion and phase movement of melatonin, We also supplemented the relevant literature, as shown in page 15.
Thank you again for your valuable comments to this article and best wishes.

Reviewer 2 Report
1. Cite a reference for PRISMA guidelines.
2. Abstract results and conclusion are confusing and not clear. Please rectify them.
3. Assessment of risk of bias and quality assessment are two different concepts in a systematic review. Authors are mixing up these two.
4. Main limitation of this review is the small sample size for each of the outcomes.
5. Provide justification for the choice of selection of the selected outcomes.
Author Response
Dear experts:
Thank you so much for your valuable comments on this article, the revision in strict accordance with the expert opinion,where the changes are made by the red font.Here are the details of the changes:
1)Following your request,we cited a reference for PRISMA guidelines,as shown in line 98.
2)According to your request, we readjusted the content in the abstract. We reviewed the relevant systematic reviews and found that most of the articles describe the literature search results in this part, and we also followed such a paradigm. We readjusted the conclusion in the abstract that exercise at different times of day may have different effects on cardiovascular risk factors in adults, but given the heterogeneity and limited number of the literature, this needs to be explored in future studies.
3)Thank you so much for pointing out our error,according to your request, we changed the literature quality assessment to the literature bias risk assessment, as shown in 2.4(lines 148-149 ,153-154).
4)We very much agree with you opinion that the main limitation of this review is the small sample size for each of the outcomes,and we added this sentence to line 24 and part 6 of the article.
5)Following your request,We explain in the introduction section why the CRP,TNF-a and IL-6 were chosen as outcome indicators of inflammatory factors(lines 41-43,46-48), We emphasized again in lines 133 – 134 that the outcome measures selected in the text are the currently widely studied mediator of exercise-promoting cardiovascular health.
Thank you again for your valuable comments to this article and best wishes.

Reviewer 3 Report
Many thanks to the editors for considering me for the review of this work and to the authors for their time. The following is a series of comments with the aim of contributing some questions that may improve the paper.
The following paper deals with a systematic review of a topic as important as a cardiovascular risk in the elderly.
In the analysis of the paper, the summary shows that it contains all the important elements of the work, although it is recommended that some parts be eliminated at the beginning to encourage rapid reading when researchers perform searches.
In the introduction, they describe the context in a very good way and highlight the most important risks, as well as the importance of physical activity to improve cardiovascular risks.
In the methodology, the authors follow the PRISMA proposal in a correct way and describe in detail the search process. Likewise, the inclusion and exclusion criteria are described in detail.
In point 2.4 it is recommended that the initials of the investigators be eliminated.
The results are clearly ordered and are interesting and useful. It is important to highlight the division by sections, which helps to understand the results and it is recommended that this structure be continued in the discussion. It is also recommended that Table 1 of the results not be left between two pages.
Author Response
Dear experts:
Thank you very much for your valuable comments on this article, the revision in strict accordance with the expert opinion,where the changes are made by the red font.Here are the details of the changes:
1) Following you request, we shortened the text in the purpose section of the abstract to make it easier and faster for readers to access the information in this article, as shown in lines 9-13.
2)We removed the initials of the researchers following you request.
3) We described in detail the literature retrieval and exclusion process(repetitive literature, animal experiments, non-English texts, and so on) following you requested, as shown in lines 162-165.
4)We separated sections in the discussion section following you request.(5.1.1,5.1.2,5.1.3,5.1.4,5.1.5)
5)Table 1 was placed on a separate page following you request.
Thank you again for your valuable comments and best wishes.

Round 2
Reviewer 2 Report
No further comments.